# An amino-terminal fragment of apolipoprotein E4 leads to behavioral deficits, increased PHF-1 immunoreactivity, and mortality in zebrafish

**Madyson M. McCarthy, Makenna J. Hardy, Saylor E. Leising, Alex M. LaFollette, Erica S. Stewart, Amelia S. Cogan, Tanya Sanghal, Katie Matteo, Jonathon C. Reeck[ID], Julia T. Oxford[ID], Troy T. Rohn[ID]***

Department of Biological Sciences, Boise State University, Boise, Idaho, United States of America

* trohn@boisestate.edu

**Data Availability Statement:** All relevant data are within the manuscript and its Supporting Information files.

## Abstract

Although the increased risk of developing sporadic Alzheimer's disease (AD) associated with the inheritance of the apolipoprotein E4 (*APOE4*) allele is well characterized, the molecular underpinnings of how ApoE4 imparts risk remains unknown. Enhanced proteolysis of the ApoE4 protein with a toxic-gain of function has been suggested and a 17 kDa amino-terminal ApoE4 fragment (nApoE4$_{1-151}$) has been identified in post-mortem human AD frontal cortex sections. Recently, we demonstrated *in vitro*, exogenous treatment of nApoE4$_{1-151}$ in BV2 microglial cells leads to uptake, trafficking to the nucleus and increased expression of genes associated with cell toxicity and inflammation. In the present study, we extend these findings to zebrafish (*Danio rerio*), an *in vivo* model system to assess the toxicity of nApoE4$_{1-151}$. Exogenous treatment of nApoE4$_{1-151}$ to 24-hour post-fertilization for 24 hours resulted in significant mortality. In addition, developmental abnormalities were observed following treatment with nApoE4$_{1-151}$ including improper folding of the hindbrain, delay in ear development, deformed yolk sac, enlarged cardiac cavity, and significantly lower heart rates. A similar nApoE3$_{1-151}$ fragment that differs by a single amino acid change (C>R) at position 112 had no effects on these parameters under identical treatment conditions. Decreased presence of pigmentation was noted for both nApoE3$_{1-151}$- and nApoE4$_{1-151}$-treated larvae compared with controls. Behaviorally, touch-evoked responses to stimulus were negatively impacted by treatment with nApoE4$_{1-151}$ but did not reach statistical significance. Additionally, triple-labeling confocal microscopy not only confirmed the nuclear localization of the nApoE4$_{1-151}$ fragment within neuronal populations following exogenous treatment, but also identified the presence of tau pathology, one of the hallmark features of AD. Collectively, these *in vivo* data demonstrating toxicity as well as sublethal effects on organ and tissue development support a novel pathophysiological function of this AD associated-risk factor.

**Funding:** This work was funded by the NIH Blueprint for Neuroscience Research 2R15AG042781-03 to Troy T. Rohn. The project described was also supported by Institutional Development Awards (IDeA) from the National Institute of General Medical Sciences of the National Institutes of Health under Grants #P20GM103408 and #P20GM109095, Directorate for Biological Sciences, 0619793, 0923535 to Julia T. Oxford. The funders had no role in study design, data collection and analysis, decision to publish, or preparation of the manuscript.

**Competing interests:** The authors have declared that no competing interests exist.

## Introduction

Alzheimer's disease (AD) is a neurodegenerative disease encompassing the most prevalent form of dementia characterized by amyloid plaques and neurofibrillary tangles (NFTs) [1, 2]. Early-onset AD has been associated with autosomal-dominant mutations in the amyloid precursor gene (APP), presenlin-1 and -2 (PSEN1/PSEN2) genes [2]. These mutations collectively comprising what is known as early-onset AD, affect approximately 5% of all known AD cases [3]. The majority of AD cases are characterized as late-onset in which the greatest risk factors for the disease are environmental (*e.g.*, aging and lifestyle choices) in addition to the inheritance of an apolipoprotein (*APOE*) allele, namely apolipoprotein E4 (*APOE4*) [2, 4]. The *APOE* gene has several isoforms of importance that are affected by a cysteine to arginine polymorphism: *APOE2* (C112, C158), *APOE3* (C112, R158), and *APOE4* (R112, R158) [5, 6]. A carrier of the *APOE4* allele increases the risk of developing AD by four- to twelve-fold [2]. However, the mechanisms of how ApoE4 contributes to increased risk of disease have remained elusive.

The *APOE* gene encodes the main cholesterol transporter protein (ApoE) in the CNS that is taken up by cells primarily through the low-density lipoprotein receptor (LDLR) family [7, 8]. Cholesterol transport in both the periphery and the CNS are vital for basal cellular function, but neurons are in critical need of adequate supply for synaptogenesis and neurite outgrowth [9]. The ApoE isoforms differ in their functional ability through the stepwise change in cysteine (C) to arginine (R) from ApoE2 to ApoE4 as described above [6]. The C112→R112 mutation has the ability to alter the side chain orientation of ApoE4 compared to ApoE2 and ApoE3 through the formation of a salt bridge combining R112 to E109 [6, 9, 10]. Data supports changes in the conformational structure of the isoforms from the C→R substitutions creates an increased likelihood of the generation of toxic fragmentation [11]. The role of ApoE4 proteolysis as a possible mechanism underlying disease risk has been supported by the findings of 17–20 kDa ApoE4 fragments in the prefrontal cortex from post-mortem AD patient tissue that localize within NFTs [11–15].

We recently examined the role of an amino-terminal fragment of ApoE4 previously identified in the human AD brain utilizing cultured BV2 microglia cells. Our findings demonstrated that exogenous application of this amino-terminal fragment of ApoE4 (nApoE4$_{1-151}$) in microglial cells resulted in uptake of nApoE4$_{1-151}$, trafficking to microglial nuclei, and the expression of numerous genes associated with inflammation [16–18]. To expand on this work, the current study employed an *in vivo* zebrafish system to study the fragment in a complex organism. The zebrafish model system has increasingly been used to study neurodegenerative diseases in vertebrates [19–21]. Some of the many benefits to this model are the rapid growth cycle, high fecundity rates, transparency of embryogenesis via externally fertilized embryos, and the early generation of stereotyped visualizable behavior at embryonic stages [22–24]. In addition, zebrafish have been utilized as a model system for studying AD including studies examining tau-induced neurodegeneration [25], neuron-glia interactions in adult fish [26] and could serve as an excellent model to facilitate potential drug discovery in AD [27].

The goal of the present study was not to employ zebrafish as a model of AD *per se*, but to test whether a risk factor associated with AD could lead to toxicity or other potential negative consequences in an *in vivo* model. In this manner, the use of wild-type zebrafish embryos was used to extend our previous *in vitro* findings in transformed cells [16–18]. Our results demonstrate that exogenous treatment of zebrafish embryos with nApoE4$_{1-151}$ led to an increase in toxicity and other morphological abnormalities. In addition, there was a trend towards decreased motor activation in the nApoE4$_{1-151}$ treated-embryos as well as enhanced PHF-1 immunoreactivity. The findings of this study suggest that the single amino acid polymorphism

from ApoE3 to ApoE4 includes a toxic gain-of-function providing a possible link between harboring the *APOE4* gene and enhance risk associated with AD.

## Materials and methods

### Synthesis of nApoE$_{1-151}$ fragments

Generation of nApoE4$_{1-151}$ and nApoE3$_{1-151}$ including synthesis of plasmid, expression in *E. coli*, and purification (>85% purity) was contracted out to GenScript Inc. (Piscataway, NJ). An anti-6X His-tag at the C-terminal end was added to facilitate purification. The verification of proteins was confirmed by DNA sequencing, SDS PAGE, and Western blot by using standard protocols for molecular weight and purity measurements. The concentration of recombinant proteins was determined by Bradford protein assay with BSA as a standard. Characterization of nApoE4$_{1-151}$ in our laboratory was assessed by ELISA and Western blot analysis utilizing an anti-His antibody as previously described [12]. A similar fragment to apoE3 (nApoE3$_{1-151}$) was utilized in order to directly compare the differences in toxicity between nApoE4$_{1-151}$ and nApoE3$_{1-151}$. Full-length, human ApoE4 protein was purchased from Prosci Inc. (Poway, CA).

### Zebrafish embryo care and maintenance

Animal husbandry and colony maintenance was handled by the Boise State University vivarium system. All animal protocols were coordinated with the Boise State University IACUC committee in accordance with recommendations from the Zebrafish Information Network (Zfin), IACUC protocol #AC18-011 and #AC21-009. Embryos were reared at 28.5˚C in an enclosed incubation unit until treatment with exogenous protein fragments. Embryos were obtained for experiments by naturally breeding adult zebrafish in breeding tanks following Boise State University established standard operating procedures. The embryos were separated from adults and cleaned prior to the start of the experiment. All embryos were cultured in a temperature-controlled incubator for the duration of embryonic use. Juvenile zebrafish and embryos were under the care of BSU animal care staff unless used for experimentation. Sacrifices of treatment subjects were made following guidance of humane endpoints. Male and female *Danio rerio* (zebrafish) were used and their approximate average weight was approximately 1.5 g each. For the tail flick assays, we justified the number of animals that were used by the following calculation (power analysis software can be found at http://www.cs.uiowa.edu/~rlenth/Power/):

For the tail-flick test (TFT): Test selection: ANOVA for number of Tail Flicks per collection time point: Embryonic Testing (0-72hpf): WT: 3 Treatment Groups, N = 150 (total embryonic samples) n = 50 (samples per treatment), SD[treatment] = 0.1568, SD[within] = 0.5, Power = 0.8. A similar calculation was used and obtained for the TEMR behavioral assay. Pain and discomfort were assessed by experiment and appropriate anesthesia (tricaine) was applied to offset and reduce pain to animals.

### Treatment of embryos with exogenous nApoE$_{1-151}$ fragments

Treatment of embryos began at 24 hpf during the 19-somite-prim-6 stage in accordance with the Kimmel Staging Series [24]. Zebrafish embryos that were staged at the time of treatment to be above 25 hours (prim-6) or below 19 hpf (20-somite stage) were excluded from experimentation. Incubation of embryos was accomplished using a mixture of E3 media (Cold Spring Harbor Protocols, recipe for E3 medium for zebrafish embryos; doi:10.1101/pdb.rec066449) with various concentrations of nApoE4$_{1-151}$ or nApoE3$_{1-151}$ protein fragments. Following staging assessments, embryos were allocated to an incubation chamber in an even distribution

depending on experiment. Embryos were kept with a density of no more than 5 embryos in a minimum of 100 μl of E3 media + protein fragment per well in 96 well plates. Control groups were raised in identical conditions with the only difference being the absence of nApoE$_{1-151}$ fragments present in E3 media. Embryos were then placed in an incubator for 24 hours of undisturbed incubation at 28.5˚C before being collected for experimentation.

### Live imaging (Light Microscope)

Live observations were recorded on an EVOS M5000 Light Microscope using brightfield settings. Embryos were imaged at 4X for a general view of the embryo(s) and 10X or greater were used for tissue specific image acquisition. Mortality was assessed as described above by comparing the number of nominal labels (1 = Alive, 0 = Dead/Non-viable) given to each group. A tally was taken via a contingency table in R to provide a raw count for each individual treatment group. Normality and equal variance were tested for prior to analysis. Each group count was then tested against each other via a Chi-square analysis and displayed via a mosaic display to indicate statistical significance by shading and examination of the Pearson residual values.

### Morphological assessments

A standardized scale was generated across multiple standards resulting in a developmental abnormality score (Fig 1). Embryos were scanned throughout the z-axis to identify internal flaws, verify pigmentation changes, and gauge organogenesis at specific stages in coordination with the Kimmel staging series as well as comparison to the non-treated controls.

### Heart rate determination

Live imaging was also applied for mortality assessments of embryos in the manner of identifying heartrate over the course of 10 second intervals. If no heart rate was detected, stimulation of embryo through water movement was performed to verify lack of response to stimuli. If no response to stimulation was found in combination with a lack of heartbeat, the embryo was deemed non-viable. Heart rate was measured in a continuous manner by beats per minute (bpm) as described above. Samples were averaged by replicate to stabilize variation within groups. Five sets of heart rate collection dates were tested through two-way ANOVA modelling with three treatment groups (Control, nApoE3$_{1-151}$, and nApoE4$_{1-151}$) in R statistical software. ANOVA model was assessed for assumptions after creation of model in

### Behavioral assays

**Tail flick behavioral test.** Larvae at 72 hpf were acclimated to the testing area for 2 minutes prior to experimentation. Five-minute videos were taken of each larvae using a Motic MGT 101 Moticam recording device with an LED-60T-B light ring. Videos were recorded down each treatment group column, then across well rows in a 96-well petri dish. Immediately following testing, the samples were euthanized using IACUC and University Guidelines, preserved in 4% PFA/PBT, then stored in 100% ethanol at -20˚C. During each recording, individual larva was documented using a numerical code to designate treatment groups for a single-blind procedure, in which one researcher recorded and encoded video names for treatment and a second researcher analyzed coded videos. Scoring was completed by examining videos while documenting the number of spontaneous tail flicks in total for each larva over a 5-minute span. A single, spontaneous tail flick was determined by counting the number of times each larva bent their tail away from center and returned to the center axis. Samples were prepared by averaging 5 tail flicks per replicate over the course of 5 replicates with both nApoE3$_{1-151}$ and

# Scale for Developmental Abnormalities

| Descriptor | Characteristics | Score | Visual Comparison |
|---|---|---|---|
| No noticeable abnormalities | •Appears to match devolpmental markers<br>•No abnormality can be visually identified<br>•Proper pigmentation displayed | 0 | |
| Minimal abnormality | •Does not meet one developmental standard<br>•No observable hindbrain folding<br>•Delay or lack of ear development<br>•Deformed yolk sac<br>•Enlarged cardiac cavity<br>•Normal Pigmentation | 1 | |
| Some abnormalities | •Does not meet two developmental standards<br>•No observable hindbrain folding<br>•Delay or lack of ear development<br>•Deformed yolk sac<br>•Enlarged cardiac cavity<br>•Craniofacial abnormality | 2 | |
| Severe abnormalities | •Does not meet three developmental standards<br>•No observable hindbrain folding<br>•Delay or lack of ear development<br>•Deformed yolk sac<br>•Enlarged cardiac cavity<br>•Craniofacial abnormality | 3 | |
| Extreme abnormality | •Meets any of the following standards:<br>-No heart rate<br>-No discernible shape<br>-Blackened or cloudy | 4 | |

**Fig 1. Semi-quantitative scale developed to assess morphological changes following treatment of zebrafish embryos with an amino-terminal fragment of nApoE4$_{1-151}$.** Rubric for developmental abnormalities was accomplished in 48 hpf zebrafish following 24-hour treatment with 25 μg/ml of nApoE41-151. This scale was established to quantify the effects of nApoE4$_{1-151}$ as compared to untreated, control embryos. Identified hallmark defects that appeared consistently following treatment with nApoE4$_{1-151}$ included inflation of pericardial cavity, enlarged hearts, pigmentation alterations, and delays or lack of development in ear and brain structures. Data are representative of 10 embryos treated with 25 μg/ml nApoE4$_{1-151}$ per trial for a total of 30 embryos.

nApoE4$_{1-151}$ at a final concentration of 25 μg/ml. Data analyzed for each trial represent the averaged results for each collection date. Data was analyzed through two-way ANOVA modeling with three treatment groups (Control, nApoE3$_{1-151}$, and nApoE4$_{1-151}$) in R statistical software. The ANOVA model was assessed for assumptions after creation of model in R.

## Touch Evoked Movement Response Assay (TEMR)

Individual larvae at 72 hpf were moved to a 14x14 mm round glass bottom petri dish filled with E3 Media 2 minutes prior to testing for acclimation under light conditions for testing. Two-minute videos were recorded on Motic MGT 101 Moticam recording device with an LED-60T-B light ring. After the start of the recording at time (T = 0), embryos were tapped lightly with a blunt probe every 15 seconds. Directly following testing, larvae were euthanized using IACUC and University Guidelines, preserved in 4% PFA/PBT, then stored in 100% ethanol at -20°C. Videos were analyzed in Noldus Behavioral Software version 15.0. Criteria for scoring was based on the number of responses following the evoked stimulus. Data was grouped by treatment and analyzed in R statistical software. Data was assessed for normality and variance. The number of responses to the evoked stimulus was analyzed using One-Way ANOVA (aov; car package). A non-parametric Kruskal-Wallis test was applied to duration data as the data was not normal.

## Immunofluorescence labeling

For all immunohistochemical procedures in this study, a standard protocol was followed for tissue preparation, and preparation of the slides. At the conclusion of treatment experiments, embryos were sacrificed prior to manual removal of chorion. Following de-chorionation, embryos were fixed in 4% paraformaldehyde (PFA)/phosphate buffered saline (PBS) overnight at 4°C and prepared for 5 μm paraffin-embedded sectioning using a Leica RM2235 Microtome at 4°C. Paraffin-embedded sections were used for staining following rehydration and a series of washes in PBS with tween-20 0.05% (PBST). Blocking of sections for non-specific staining was accomplished using a standard incubation buffer consisting of 1% normal goat serum, 2% bovine serum albumin in PBST for 2 hours at room temperature. Primary antibodies, detailed in **Table 1**, were incubated with slides for 18–24 hours at 4°C. Sections were then washed in triplicate for 5 minutes with PBST. Slides were then incubated in the appropriate secondary antibody for 1 hour at room temperature. Following the final wash, DAPI infused soft mount was placed on slides and allowed to set before coverslip addition. Sections were analyzed for preliminary staining with EVOS M5000 light cubes at lowest intensity settings. Sections were then kept in the dark at 4°C until used for confocal imaging. Following labeling, confocal assessment of the localization of nApoE4$_{1-151}$ in neuronal cell populations was as previously described [18]. All images and z-stacks generated were obtained using Zeiss Microscope, LSM 510 Meta confocal imaging system (Carl Zeiss, Oberkochen, Germany) and processed using Zen blue edition (Carl Zeiss, Göttingen, Germany). For each area of interest, a minimum of 3 sections per embryo were stained.

## Results

We chose to examine the role of this specific amino-terminal fragment of nApoE4$_{1-151}$ for several reasons. First, we documented widespread evidence for this fragment in the human AD

**Table 1. Description of antibodies used for immunofluorescence experiments.**

| Primary Antibodies | Source | Secondary Antibodies | Recognition |
|---|---|---|---|
| Anti-6X His Tag (1:500, Rabbit polyclonal) | Abcam, Inc. | AF 488 (1:500, Goat, Anti-Rabbit) | Reacts specifically with His-tagged ApoE3 or E4 fragments |
| NeuN (1:50, Mouse monoclonal) | Abcam, Inc. (1B7) | AF 555 (1:200, Donkey, Anti-Mouse) | Recombinant full-length NeuN |
| PHF-1 (1:250, Mouse polyclonal) | Dr. Peter Davies (Albert Einstein College of Medicine, Bronx, NY) | AF 555 1:200, Donkey, Anti-Mouse) | Serine-396 and serine-404 phosphorylated sites of tau |

brain, where it localized within nuclei of microglia and neurons [18]. Second, generation of this fragment was documented following incubation of full-length ApoE4 with matrix metallo-proteinase-9 (MMP-9) [18]. Finally, recent data from our lab suggests that, *in vitro*, nApoE4$_{1-151}$ can traffic to the nucleus leading to toxicity and expression of inflammatory genes in BV2 microglia cells [16, 17, 28]. It is noteworthy that because zebrafish embryos do not yet express fully functional glial cells at 24 hpf including microglia [29], our study focused on the effects of nApoE4$_{1-151}$ on neuronal populations. We previously identified the nApoE4$_{1-151}$ fragment within neurons of the human AD brain [18]. The purpose of this current study is to expand those findings *in vivo*, by assessing the mechanisms by which this fragment may induce toxicity, developmental abnormalities, and behavior deficits in a model system consisting of zebrafish.

## Morphological assessments

A semi-quantitative morphological assessment revealed treatment group specific phenotypes that were predominantly present in the nApoE4$_{1-151}$-treatment group. A scale was generated across multiple standards resulting in a developmental abnormality score (Fig 1).

Embryos were scanned throughout the z-axis to identify internal flaws, verify pigmentation changes, and gauge organogenesis at specific stages in coordination with the Kimmel staging series as well as comparison to the non-treated controls. Control groups were used to set a standard of comparison of which nApoE3$_{1-151}$ followed closely in most regards except for loss of pigmentation in both treated groups (blue arrows, Fig 2A–2C). Features most commonly observed for nApoE4$_{1-151}$ groups in addition to loss of pigmentation were lack of hindbrain folding at cerebellar primordium (orange arrows, Fig 2A–2C), as well as enlargement of the cardiac cavity (red arrow, Fig 2C). Quantification of morphological abnormalities are depicted in Fig 2D based on our standardized scale, with nApoE4$_{1-151}$ showing the most severe degree of changes following treatment with a significant increase in developmental abnormality scores compared to untreated controls. Heart rate measurements reported a significant decrease in the average heart rate compared to untreated controls at both low (25 μg/ml) and high concentrations of nApoE4$_{1-151}$ (50 μg/ml) (Fig 2E).

## Survivability and mortality assessments

Survivability curves indicated that treatment of zebrafish embryos at 24 hpf lead to significant mortality. There was a 90–100% reduction in viable embryos 48 hours after treatment began (Fig 3A). In addition, nApoE4$_{1-151}$ treated-embryos failed to recover after removal of treatment media. The survivability of embryos following treatment with 25 μg/ml of nApoE4$_{1-151}$ was reduced by 50% within 24 hours of treatment (yellow dotted line), whereas at 50 μg/ml, it was reduced by 50% by 12 hours (green dotted line, Fig 3A). As a control, we also tested the impact of nApoE3$_{1-151}$ on survival. In this case, identical concentrations of nApoE3$_{1-151}$ had little impact on embryo survival following treatment even at the highest concentration of 50 μg/ml (<15%) (blue dashed line, Fig 3A).

Mortality data reported a high nApoE4$_{1-151}$ associated mortality compared to both nApoE3$_{1-151}$ and non-treatment groups (p-value compared to non-treated controls was $1.17e^{-13}$) (Fig 3B). nApoE3$_{1-151}$ and controls revealed a nearly identical mortality rate with <10% drop in mortality for nApoE3$_{1-151}$ compared to untreated controls. These results suggest that changing a single amino acid (C>R) at position 112 is sufficient to induce significant toxicity when zebrafish embryos are treated exogenously with nApoE4$_{1-151}$. We also tested the ability of human, full-length ApoE4 to induce mortality using the highest concentration tested for nApoE4$_{1-151}$ (50 μg/ml). As shown in S1D Fig, full-length ApoE4 had minimal effect on

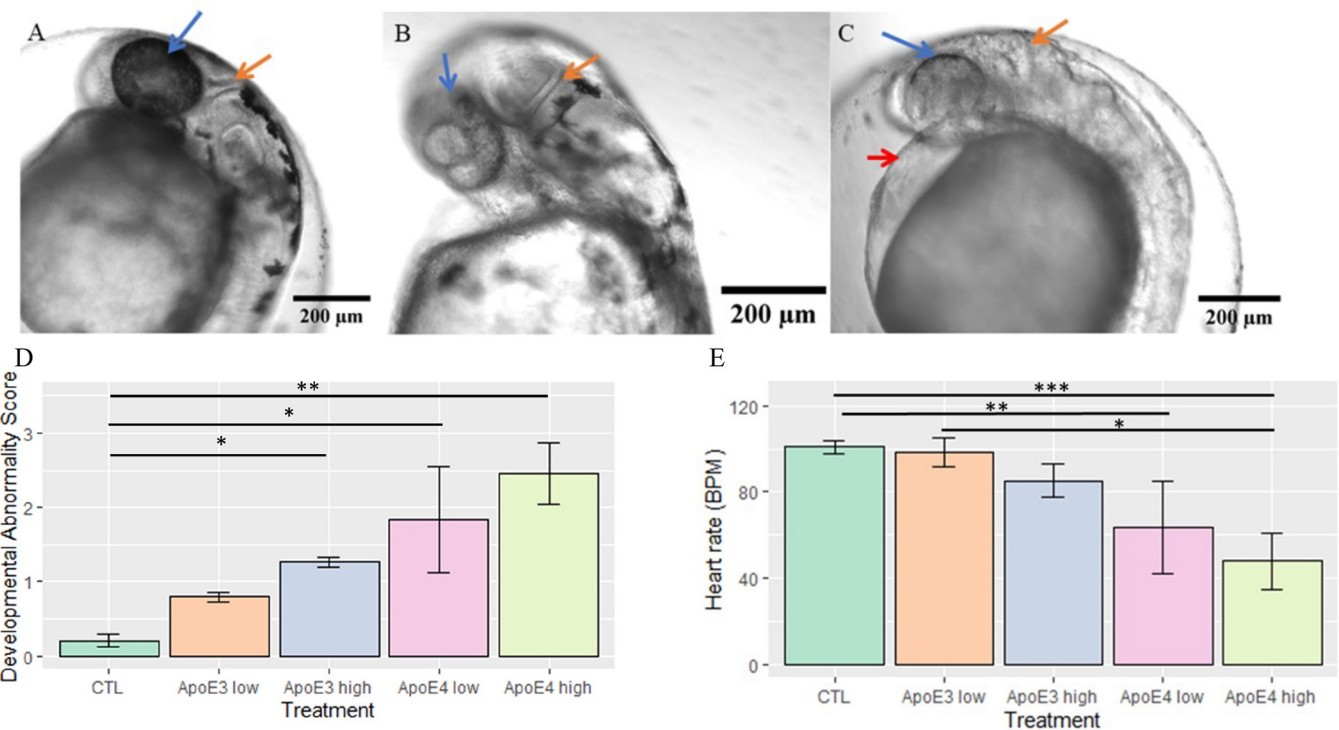

**Fig 2. A sublethal concentration of nApoE4$_{1-151}$ leads to morphological abnormalities in zebrafish embryos at the hatching phase.** Representative light phase contrast microscopic images following live imaging of embryos at 48 hpf following a 24-hour period with respective treatments (Control, 25 μg/ml of nApoE3$_{1-151}$, or 25 μg/ml of nApoE4$_{1-151}$. **A:** Arrows point to consistent morphological changes as compared to untreated controls that are healthy and categorized as having a developmental abnormality score of <1.0 (Panel A). The blue arrows designate pigmentation changes; orange arrows designate cerebellar primordium junction differences in the hindbrain. **B:** Exogenous treatment with nApoE3$_{1-151}$ impacted pigmentation pattern in otherwise healthy embryos. Embryos in this category were most likely to receive a score of <1. **C:** 24-hour incubation of a sublethal concentration of nApoE4$_{1-151}$ resulted in developmental abnormality scores >3. Embryos in this category were typically observed to be delayed in development with limited hindbrain folding (orange arrow), limited or lacking pigmentation (blue arrow), as well as enlargement of the cardiac cavity (red arrow). **D.** Quantitative developmental abnormality scores for each treatment group following treatment of embryos for 24 hours with respective fragments at low E3 (orange bar), E4 (pink bar) concentrations (25 μg/ml) or at high concentrations (50 μg/ml) E3 (gray bar), E4 (yellow bar). The nApoE4$_{1-151}$ 25 μg/ml-treated groups were significantly different from controls (H(4) = -2.43, $p$ = 0.0074). At 50 μg/ml both nApoE4$_{1-151}$ (H(4) = -3.32, p = 0.0004) and nApoE3$_{1-151}$-treatment groups (H(4) = -1.77, p = 0.037) were significantly different from controls. Errors bars represent ± S.E.M. *p<0.05, **p<0.01, ***p<0.001 **E.** Heart rate data obtained from live microscope analyses in 25 μg/ml and 50 μg/ml treatment groups nApoE3$_{1-151}$ and nApoE4$_{1-151}$ compared to non-treated controls. nApoE4$_{1-151}$ (pink bar, 25 μg/ml) was significantly different from controls (H(4) = 1.77, $p$ = 0.038). nApoE4$_{1-151}$ (yellow bar, 50 μg/ml) was significantly different from nApoE3$_{1-151}$ 25 μg/ml (H(4) = 1.94, p = 0.026) and controls (H(4) = 2.665, p = 0.0036). Errors bars represent ±S.E.M. All other comparisons were insignificant. *p<0.05, **p<0.01, ***p<0.001.

toxicity and data were not significant different from non-treated controls. In contrast, treatment with nApoE4$_{1-151}$ lead to significant mortality (>90%, S1D Fig) as well as severe morphological damage (S1A–S1C Fig). Interesting, in non-treated controls all fish were healthy and alive but remained in their chorion (15/15 embryos, S1A Fig). For full-length ApoE4, 13/15 embryos were out of their chorion (87%) in three independent experiments. This data suggest that not only is full-length ApoE4 non-toxic, but that it actually accelerates hatching of zebrafish larvae.

## Nuclear localization of nApoE4$_{1-151}$ and the presence of tau pathology

To assess the subcellular localization of nApoE4$_{1-151}$ following treatment of zebrafish embryos, confocal analysis was undertaken following fixation and sectioning of treated embryos. To track potential nApoE4$_{1-151}$ uptake, we utilized an anti-6X His-tag antibody (1:500), together with DAPI (nuclear stain) and markers for neuronal cells including Neuronal Nuclei (NeuN).

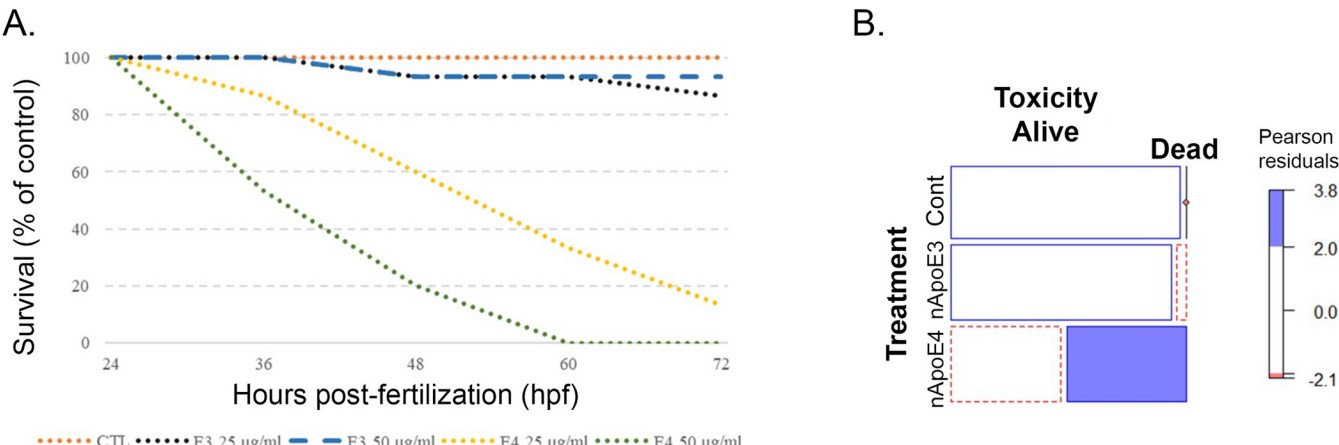

**Fig 3. Survivability is decreased in zebrafish embryos following exogenous treatment with an amino-terminal fragment of nApoE4$_{1-151}$. A.** Embryos at 24 hpf (prim-9 stage) were segregated into three groups: controls (untreated), 25 μg/ml or 50 μg/ml nApoE3$_{1-151}$, and 25 μg/ml or 50 μg/ml nApoE4$_{1-151}$. Embryos that lacked a heartbeat for 10 seconds were stimulated to induce movement. If no movement or heartbeat was detected, embryos were considered to be non-viable. Significant mortality was observed at both concentrations of nApoE4$_{1-151}$ by 48 hpf (orange and green dotted lines) compared to non-treated controls (orange dotted line) or nApoE31-151 (blue dashed line and black dotted lines). N = 3 independent trials, N = 5 fish/treatment. **B.** The Mosaic Plot depicts mortality based on a lack of heartbeat and response to physical stimuli following exogenous treatment of 48 hpf zebrafish embryos with either 25 μg/ml nApoE3$_{1-151}$ or nApoE4$_{1-151}$ for 24 hours. The blue filled region of the bar graph designates 25 μg/ml treatment of nApoE4$_{1-151}$ which led to a significant portion of the embryos being designated as dead. The red dotted portion of the bar graphs indicates less than expected were alive (p = 1.17e-13). All blank cells indicate the sample group followed the estimated trend. Data indicated significant morality for only the nApoE4$_{1-151}$ group. N = 3 independent experiments, 15 embryos/treatment.

NeuN is part of the RNA splicing machinery and is predominantly found in the nucleus of post-mitotic neurons [30]. As depicted in Fig 4B, nApoE4$_{1-151}$ is taken up by neurons following exogenous treatment of 48 hpf zebrafish embryos that colocalized with NeuN (Fig 4B and 4C). Of interest was the punctate staining of nApoE4$_{1-151}$ which supports our previous staining pattern observed in transformed BV2 microglial cells, suggesting possible aggregation of the nApoE4$_{1-151}$ fragment [18]. Also displayed in Fig 4 is staining from a representative, whole embryo mount taken at 10X magnification. Staining of nApoE4$_{1-151}$ is noted throughout the nervous system (green labeling, Fig 4D). Strong co-localization between nApoE4$_{1-151}$ and NeuN was noted in the region of the medulla (arrows, Fig 4D).

Additional immunofluorescence studies were undertaken to assess any potential relevance to known AD pathology. Previous studies have demonstrated that amino-terminal fragments of ApoE4 localize in NFTs of the human AD brain [12] and may induce neurofibrillary changes in cultured neurons [11]. Zebrafish are known to express gen orthologues to the human *MAPT* gene including *MAPTA* and *MAPTB* [31]. Therefore, we examined whether treatment of zebrafish embryos with nApoE4$_{1-151}$ led to similar pathological changes to tau by confocal IF using PHF-1 that recognizes hyperphosphorylated, fibrillar forms of tau present in the human AD brain. For these experiments we used embryos at 72 hpf due to our preliminary findings that staining of PHF-1 in 48 hpf was relatively weak following treatment. Robust labeling of PHF-1 within apparent neurons was evident under these experimental conditions (arrows, Fig 5I–5L). In contrast, little PHF-1 labeling was observed with nApoE3$_{1-151}$ under identical experimental conditions (Fig 5E–5H). In Fig 5K, a merged, high-magnification of another representative treated embryo is shown. In this case, PHF-1 labeling appeared fibrillar in nature, which is characteristic of PHF-1 staining within NFTs of the AD brain [32]. These results support the linkage of nApoE4$_{1-151}$ to one of the significant hallmark pathologies found in AD, namely neurofibrillary tangles.

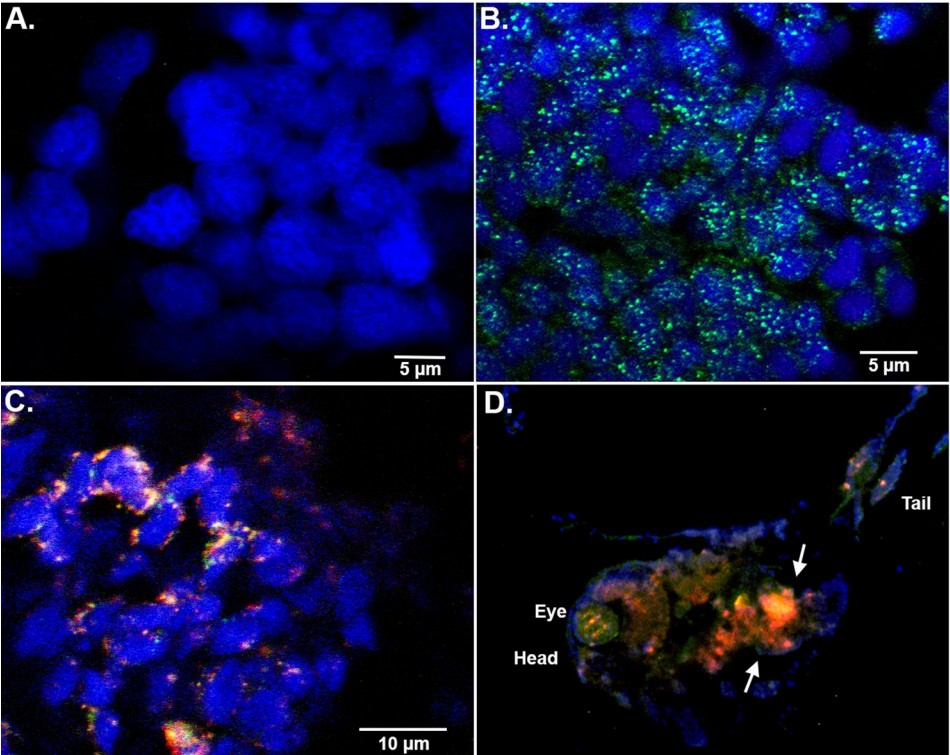

**Fig 4. Exogenous treatment of zebrafish embryos with nApoE4$_{1-151}$ leads to nuclear localization. A-C.** Representative images from confocal immunofluorescence in 5 mm paraffin-embedded sections that were stained with DAPI (A), anti-His antibody (green, B), or anti-His together with NeuN (C). There was no detection of nApoE4$_{1-151}$ fragments in untreated control neuronal cells as indicated by the lack of labeling in Panel A. Nuclear localization of the nApoE4 fragment was evident (Panels B and C) following exogenous treatment. For the E4 fragment, staining appears punctate and co-localized with NeuN and DAPI (B and C). All images were captured within the area of the cerebellum and fourth ventricle. **D.** Identical to Panels B-C with the exception that whole embryo mounts were triple labeled in order to display overall labeling in the entire organism at low magnification. Labeling of head, eye and tail is presented for orientation. The intense orange fluorescence area (arrows) represents regions with strong overlap between the E4 fragment and NeuN. In this case, labeling of the E4 fragment that co-localized with DAPI and NeuN was apparent in the hindbrain brain region. Data are representative of five independent experiments.

## Motor deficits in juvenile zebrafish following treatment with a sublethal concentration of nApoE4151

As an initial approach, we assessed whether a stereotypical motor behavior in zebrafish, spontaneous tail flicking, was diminished following treatment with nApoE4$_{1-151}$ or nApoE3$_{1-151}$. For these experiments only fish that were deemed viable and without noticeable, severe morphological abnormalities were utilized. Fig 6A depicts the results of this experiment showing that zebrafish treated with nApoE4$_{1-151}$ performed the least number of tail flicks per group with >50% reduction from untreated controls. Treatment of zebrafish larvae with nApoE3$_{1-151}$ performed similarly to controls. However, no difference was significant across any group ($F$ (2,27) = 1.24, $p$ = 0.305). It's noteworthy that control zebrafish demonstrated minimal spontaneous tail flick behavior and there were large variations within groups. The lack of response in even our control cohorts indicates that there is limited motor movement in 72 hpf in an unstimulated environment. Therefore, a second motor behavior task was undertaken whereby zebrafish were stimulated to move using a blunt instrument. This type of behavior is known as the touch-evoked movement response (TEMR) test and results mirrored the findings from the spontaneous tail-flick experiment (Fig 6B). In this case, control and nApoE3151-treated

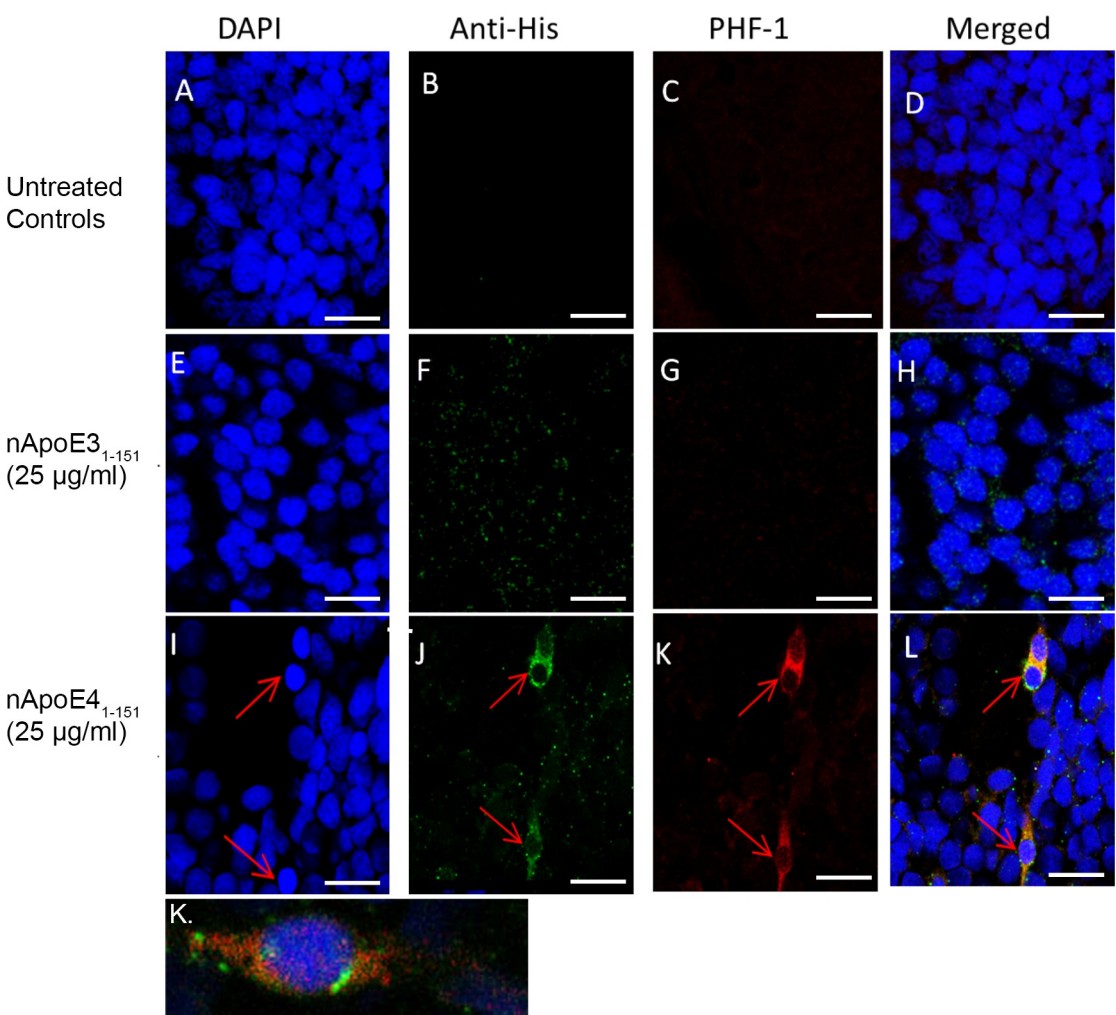

**Fig 5. Tau pathology present after treatment with exogenous nApoE4₁₋₁₅₁ fragment in 72 hpf zebrafish brain.** Representative 40X images from confocal immunofluorescence in 5 mm paraffin embedded sections of non-treated control 72 hpf zebrafish (A-D), nApoE3₁₋₁₅₁-treated at 25 μg/ml (E-H), or nApoE4₁₋₁₅₁-treated at 25 μg/ml (I-L). Strong PHF-1 labeling was only observed following treatment with nApoE4₁₋₁₅₁ (I-L). Panel K depicts a separate, representative merged image following treatment with nApoE4₁₋₁₅₁. In this case, at high magnification the fibrillar nature of PHF-1 labeling was apparent. All scale bars represent 50 μm. Data are representative of three independent experiments.

groups responded to 90% of applied stimulations. In contrast, nApoE4151-treated fish responded to fewer than 50% of evoked stimulations (Sup1-3 video files). However, as in Fig 6A, due to high variations within groups, no statistical significance was observed (F(12) = 1.482, p = 0.266). We also assessed the total distance traveled as well as the cumulative duration of response during the TEMR test and in this case both nApoE3 and E4 groups showed a downward trend although similar to the previous findings, no significant differences (p-value <0.05) between groups were observed (Fig 6C and 6D). Taken together, our results show strong trends for motor behavior impairments following treatment of zebrafish with sub-toxic concentrations of nApoE4151, that were not present in non-treated controls.

Due to the observations that tail flick behavior was abnormal following treatment of embryos with nApoE4₁₋₁₅₁, we examined PHF-1 and nApoE4₁₋₁₅₁ staining in the developing

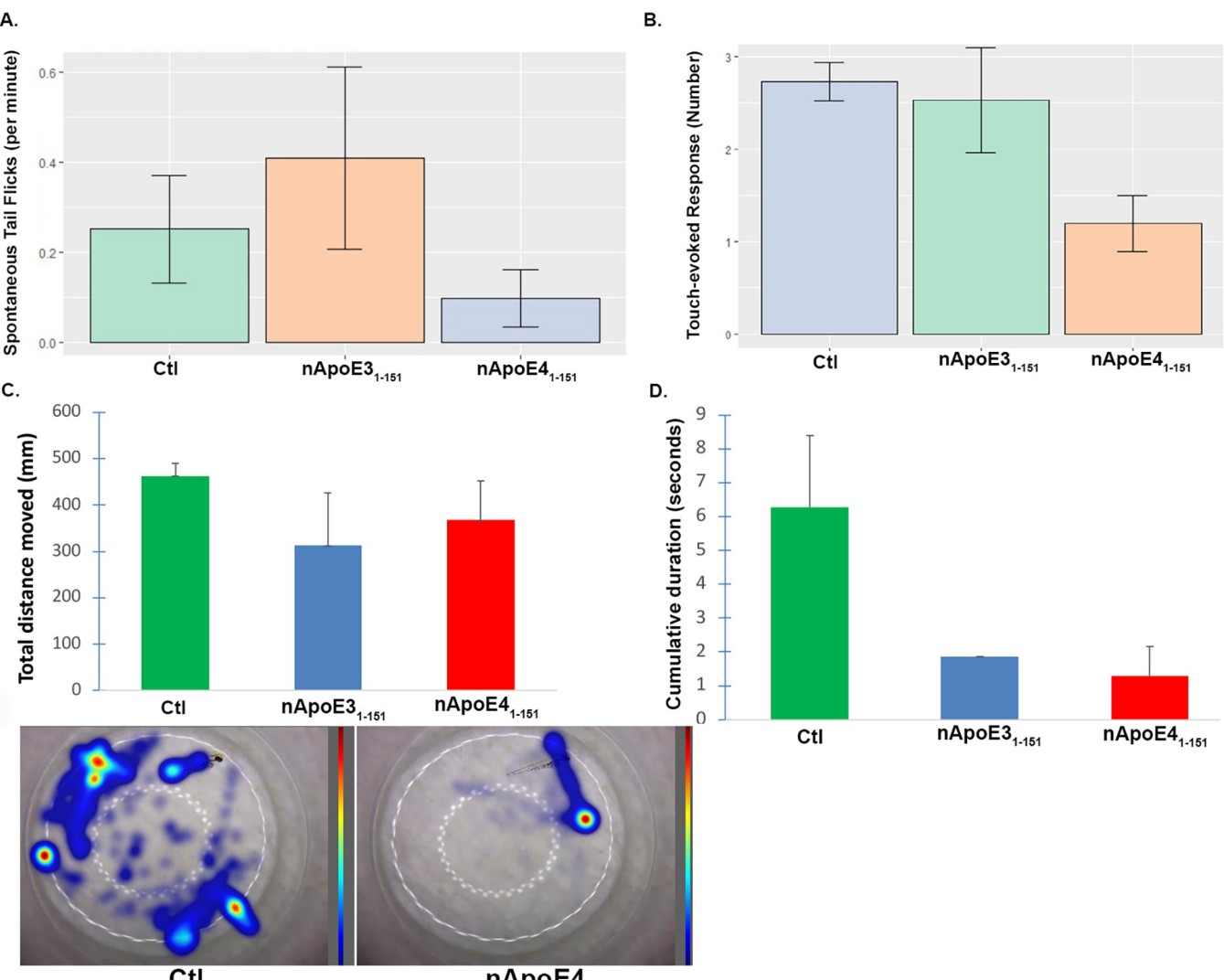

**Fig 6. Negative trends in motor behavior in zebrafish following treatment with nApoE4$_{1-151}$. A.** Groups for non-treated controls (green bar), nApoE3$_{1-151}$ 25 μg/ml (orange bar), and ApoE4$_{1-151}$ 25 μg/ml (blue bar)) were assessed via video monitoring to determine number of spontaneous tail flicks per minute that were then averaged per group for each trial. Data are representative of N = 5 trials, for a total of 25 embryos per group, ±SEM. Data depicted show limited spontaneous tail flick activation from every group with no difference detectable between groups ($F(2,27) = 1.24$, $p = 0.305$). **B.** Results from the touch-evoked response motor behavior experiment. Non-treated controls had a 90% response rate to the evoked, tactile stimulus, whereas for nApoE4$_{1-151}$-treated groups responded to fewer than 50% of stimuli. No significant difference was observed ($F(12) = 1.482$, p = 0.266). **C and D.** Results from TEMR analyses similar to Panels A and B with the exception that in this case, total distance traveled (C) or the total time swimming (D) were recorded via video monitoring and using Noldus tracking software. For Panel C, representative heat maps of individual larvae representing either wild-type controls (left Panel), or a low-performing nApoE4$_{1-151}$-treated zebrafish (right Panel). Each bar represents the average total distance traveled or averaged cumulative duration during 5 independent trials, for a total of 15 embryos per group, (±SEM). No significant differences were observed, with for example the Ctl group vs. nApoE4$_{1-151}$ having a p value = 0.08 in Panel C. P-values for Panel D were Ctl vs. E3 fragment = 0.86 and Ctl vs. E4 fragment = 0.06.

tail region by confocal microscopy. As shown in Fig 7, typical punctate nApoE4$_{1-151}$ labeling was observed in apparent skeletal muscle cells (Fig 7E). However, in this case, we were unable to identify nuclear localization of the fragment within muscle cell populations. Instead, it appeared most labeling was cytoplasmic (Fig 7F). In nApoE4$_{1-151}$-treated embryos we observed skeletal muscle formations that appeared disorganized with cells exhibiting abnormal morphologies (Fig 7E). These data could explain the motor behavioral deficits even following removal of treatment media.

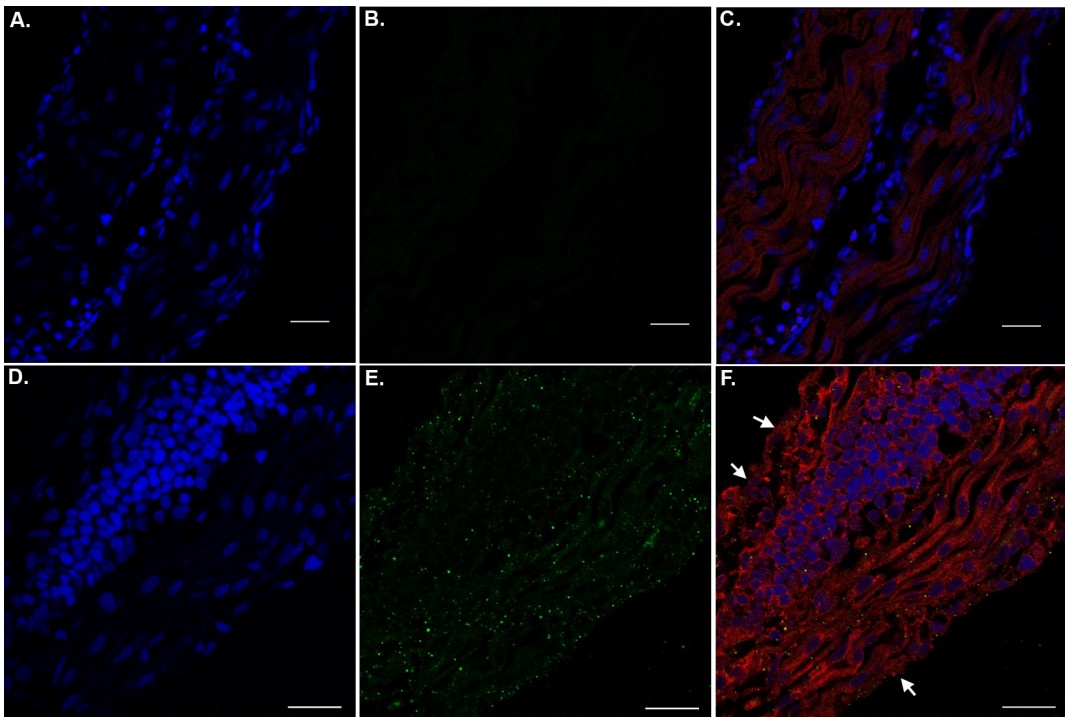

**Fig 7. The presence of nApoE4$_{1-151}$ within tail regions of zebrafish embryos. A-C:** Representative images from confocal immunofluorescence in 5 mm paraffin-embedded sections of non-treated control 48 hpf zebrafish embryos that were stained with DAPI (A), anti-His antibody (1:500) (B), and the merged image together with PHF-1 (1:250) in Panel (C). There was no detection of any nApoE4$_{1-151}$ fragments in untreated control sections as indicated by the lack of labeling in Panel B. **D-F:** Identical to Panels A-D with the exception that embryos were exogenously treated for 24 hours with 25 μg/ml of nApoE4$_{1-151}$. In this case punctate staining of the fragment was observed that appeared cytoplasmic. PHF-1 labeling was identified in muscle cells that exhibited abnormal morphology (arrows, Panel F). All scale bars represent 20 μm.

## Discussion

Despite intensive research efforts, the pathophysiological relationship between harboring the *APOE4* allele and the development of late-onset AD remains largely unknown. This question is further complicated by the fact that only the ApoE4 protein represents a significant risk factor even though it differs from ApoE3 by a single amino acid at position 112 and ApoE2 by two amino acids (positions 112 and 158) [6]. Both substitutions lead to the replacement of a cysteine residue with an arginine residue [6]. One possible hypothesis leading to increased dementia risk is the propensity of the ApoE4 isoform to be highly susceptible to proteolysis compared to E3 and E2 [33]. Prior research from our lab supports the hypothesis that ApoE4 fragmentation via the metalloproteinase-9 (MMP-9) may contribute to AD pathology and inflammation [18]. Additional studies have also shown neurotoxic and pro-inflammatory responses to the fragmentation of ApoE4 [34–37]. Recently we identified a 151 amino-terminal fragment of ApoE4 (nApoE4$_{1-151}$) that localized within the nucleus of both neurons and microglia cells of the human AD brain [18] and *in vitro*, we demonstrated this fragment is taken up by BV2 microglia cells, traffics to the nucleus and leads to the upregulation of thousands of genes, many of which associated with microglia activation and inflammation [16, 28].

In the current studies, we expanded these findings utilizing an *in vivo* model system consisting of zebrafish. The primary weakness in our previous published work was the use of transformed, BV2 microglia cells in an entirely *in vitro* model system. These weaknesses can be summarized as relying on data from a single, murine, immortal microglia cells that may not be

representative of normal, non-transformed cells. Another potential caveat of our previous studies was the reliant upon *in vitro* model systems to investigate the pathophysiological actions of nApoE4$_{1-151}$. Therefore, the primary goal of the current study was to expand our *in vitro* findings to an intact organism consisting of zebrafish embryos and larvae. Zebrafish have emerged an excellent model organism for studies of vertebrate biology. One of the more distinct advantages of the zebrafish is the optical clarity of the embryos allowing for the investigation throughout the developmental process using non-invasive imaging techniques. There is also a high degree of conservation between zebrafish and human brain organization including both neuroanatomic [38, 39], neurochemical [40], and behavioral circuitry including learning [41], touch [42], and decision making [43]. Moreover, zebrafish have served as excellent models to study the pathophysiology underlying AD including a study that demonstrated intraventricular injection of Aβ$_{1-42}$ in the embryonic brain leads to memory loss and cognitive deficits along with increased tau phosphorylation [44, 45]. Taken together, the zebrafish presents itself as a novel model system to examine the potential effects of nApoE4$_{1-151}$ and we choose to examine wild-type zebrafish only so that the potential effects of nApoE4$_{1-151}$ could be examined singularly without any potential confounding variables that could arise in mutant zebrafish harboring APP or tau mutations.

As an initial approach, we treated zebrafish embryos at 24 hpf and examined any potential morphological changes 24 hours later (48 hpf). Compared to untreated controls, nApoE4$_{1-151}$ exposure produced significant toxicity and led to changes to the nervous system, the heart, and loss of pigmentation. The nervous system was visibly impacted by the lack of hindbrain folding that is typical in 48 hpf zebrafish along the cerebellar primordium. Additionally, nApoE4$_{1-151}$ treatments at both a low (25 μg/ml) and high (50 μg/ml) concentrations produced significant differences from controls in terms of developmental abnormalities and reduced heart rates. The morphological results of enlargement of hearts following nApoE4$_{1-151}$ treatment are intriguing based on the well-known link between inheritance of *APOE4* and an increased risk of cardiovascular disease [46]. Confocal imaging revealed the colocalization of nApoE4$_{1-151}$ with NeuN within the hindbrain region around the medulla oblongata which regulates heart rate. These data support the hypothesis that nApoE4$_{1-151}$ may destabilize the hindbrain networks during the incubation period leading to downstream cardiovascular deficits, including a reduced heart rate.

From early stages of development, zebrafish swimming behavior and response to external stimuli can be assessed. The responses to external stimuli can be detected at early larvae states (72 hpf) in which zebrafish show escape response swimming behavior, for example in response to touch directed to either the head or tail [47]. Similar to the cardiovascular system, the touch-evoked response is under control of the hindbrain [48, 49]. Specifically, the touch escape response and spontaneous tail flicks are both proposed to be regulated by the Mauthner cells located in the zebrafish hindbrain [50, 51]. Treatment of zebrafish larvae with nApoE4$_{1-151}$ led to less than half of the stimulation attempts whereas non-treated controls and nApoE3$_{1-151}$ responded to over 90% and 80% stimulation attempts, respectively. Downward trends for both total distances swam, and duration times were also observed following treatment with nApoE4$_{1-151}$, that just missed statistical significance. Increased detection of nApoE4$_{1-151}$ compared to controls within the hindbrain region could be the rationale for the effects observed in both of these locomotor assays. Presently, it is not known how nApoE4$_{1-151}$ leads to these motor deficits or at what stage of this behavior does nApoE4$_{1-151}$ interfere: initiation or execution of the behavioral response? Initiation of the response would imply the sensation of the stimulus was never received through the rohon-beard cells or the dorsal root ganglia which have been shown to share the responsibility of tactile sensation during the 72 hpf period [52].

Another finding in the present study was enhanced mortality induced by exogenous treatment of nApoE4$_{1-151}$, which was not observed in parallel experiments with nApoE3$_{1-151}$ nor following treatment with full-length, human ApoE4. These results support our previous *in vitro* findings indicating enhanced cellular toxicity of only nApoE4$_{1-151}$, which differs by a single amino acid at position 112 (R>C) [18]. The further identification of nApoE4$_{1-151,}$ within the nucleus of neurons of the developing nervous system of zebrafish supports the hypothesis that the uptake of this fragment and trafficking to the nucleus may lead to stimulation of cell death pathways similar to what we have recently observed in BV2 microglia cells [16, 28].

In the context of AD, an important finding of our results was the evolution of tau pathology following exogenous treatment of zebrafish with nApoE4$_{1-151}$ at 72 hpf. These findings support that nApoE4$_{1-151}$ may promote tau pathology, a hallmark feature seen in human AD pathology [53]. Previous studies have linked the presence of amino-terminal fragments of ApoE4 with NFTs in the human AD brain as well as in various animal models including following expression of mutant forms of tau in zebrafish [15, 25, 33]. The observed tau pathology can lead to disruptions in axonal function which in turn, can have deleterious effects on neuronal signaling and axonal transport [54, 55].

A final observation was staining within skeletal muscle located within the tail region of developing zebrafish (Fig 7). We observed skeletal muscle formations that appeared disorganized with cells exhibiting abnormal morphologies. These data could explain the motor behavioral deficits even following removal of treatment media. Our results are similar in terms of muscle cell disorganization observed in zebrafish expressing mutant acetylcholine receptors at the neuromuscular junction [56]. Further experiments will be necessary to fully understand how nApoE4$_{1-151}$ may be exerting this effect in skeletal muscle.

## Conclusion

The *APOE4* allele stands out as the greatest risk factor for late-onset AD, as *APOE4* carriers account for 65–80% of all cases [57]. Although ApoE4 plays a normal role in lipoprotein transport, how it contributes to AD pathogenesis remains speculative. Recent data from our lab suggests that, *in vitro*, nApoE4$_{1-151}$ can traffic to the nucleus leading to toxicity and expression of inflammatory genes in BV2 microglia cells [16, 18, 28]. In the present study, we now expand those findings *in vivo*, by demonstrating toxicity of nApoE4$_{1-151}$, and motor behavior deficits in a novel model system consisting of zebrafish. Taken together, these results support the hypothesis that a key step in mechanism of action is the cleavage of full-length ApoE4, generating an amino-terminal fragment that exhibits a toxic-gain of function. Therefore, the neutralization of this amino-terminal fragment of ApoE4, specifically, may serve as an important therapeutic strategy in the treatment of AD.

## Supporting information

**S1 Fig. Full-length ApoE4 does not induce toxicity in zebrafish embryos.** Representative images displaying morphological effects following treatment with either 50 μg/ml full-length ApoE4 (B) or nApoE4$_{1-151}$ (C) indicated that full-length ApoE4 showed little effects on morphology as compared to non-treated controls (A). Embryos treated with full-length ApoE4 resulted in 13/15 fish had exiting their chorion. In contrast, significant morphological effects (arrow, C) and mortality were observed in the nApoE4$_{1-151}$-treatment group (D). Data are representative of three independent experiments (N = 5 per group). No significant difference was observed in mortality between non-treated controls and full-length ApoE4 (p = 0.19). Significant mortality (93%) was observed in the nApoE4$_{1-151}$-treatment group (p = .000076). (TIF)

**S1 Video. Raw video file depicting a representative wild-type zebrafish recorded during the TEMR behavior test.** Two-minute videos were recorded on Motic MGT 101 Moticam recording device with an LED-60T-B light ring. After the start of the recording at time (T = 0), embryos were tapped lightly with a blunt probe every 15 seconds. See details in materials and methods section.
(MP4)

**S2 Video. Representative raw video file depicting a representative nApoE4$_{1-151}$-treated zebrafish recorded during the TEMR behavior test.** Two-minute videos were recorded on Motic MGT 101 Moticam recording device with an LED-60T-B light ring. After the start of the recording at time (T = 0), embryos were tapped lightly with a blunt probe every 15 seconds. See details in materials and methods section.
(MP4)

**S3 Video. Representative raw video file depicting a representative nApoE4$_{1-151}$-treated zebrafish recorded during the TEMR behavior test.** Two-minute videos were recorded on Motic MGT 101 Moticam recording device with an LED-60T-B light ring. After the start of the recording at time (T = 0), embryos were tapped lightly with a blunt probe every 15 seconds. See details in materials and methods section.
(MP4)

## Author Contributions

**Conceptualization:** Madyson M. McCarthy, Saylor E. Leising, Amelia S. Cogan, Jonathon C. Reeck, Julia T. Oxford, Troy T. Rohn.

**Data curation:** Madyson M. McCarthy, Makenna J. Hardy, Saylor E. Leising, Erica S. Stewart, Amelia S. Cogan, Troy T. Rohn.

**Formal analysis:** Madyson M. McCarthy, Makenna J. Hardy, Saylor E. Leising, Erica S. Stewart, Amelia S. Cogan, Katie Matteo, Jonathon C. Reeck.

**Funding acquisition:** Julia T. Oxford, Troy T. Rohn.

**Investigation:** Makenna J. Hardy, Saylor E. Leising, Alex M. LaFollette, Erica S. Stewart, Amelia S. Cogan, Tanya Sanghal, Katie Matteo.

**Methodology:** Madyson M. McCarthy, Jonathon C. Reeck.

**Project administration:** Julia T. Oxford, Troy T. Rohn.

**Resources:** Julia T. Oxford, Troy T. Rohn.

**Supervision:** Jonathon C. Reeck, Julia T. Oxford, Troy T. Rohn.

**Validation:** Makenna J. Hardy.

**Visualization:** Makenna J. Hardy, Saylor E. Leising.

**Writing – original draft:** Madyson M. McCarthy, Makenna J. Hardy, Troy T. Rohn.

**Writing – review & editing:** Troy T. Rohn.

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
