## [Decision Letter · Decision Letter 0]

15 Sep 2022

PONE-D-22-19048An amino-terminal fragment of apolipoprotein E4 leads to behavioral deficits, increased PHF-1 immunoreactivity, and mortality in zebrafishPLOS ONE

Dear Dr. Rohn,

Thank you for submitting your manuscript to PLOS ONE. After careful consideration, we feel that it has merit but does not fully meet PLOS ONE’s publication criteria as it currently stands. Therefore, we invite you to submit a revised version of the manuscript that addresses the points raised during the review process.

We look forward to receiving your revised manuscript.

Kind regards,

Weidong Le

Academic Editor

PLOS ONE

Journal Requirements:

"This work was funded by the National Institutes of Health Grant 2R15AG042781-02A1. The project described was supported by Institutional Development Awards (IDeA) from the National Institute of General Medical Sciences of the National Institutes of Health under Grants #P20GM103408 and #P20GM109095. We also acknowledge support from the Biomolecular Research Center at Boise State with funding from the National Science Foundation, Grants #0619793 and #0923535; the M.J. Murdock Charitable Trust; and the Idaho State Board of Education."

5. Please include captions for your Supporting Information video files at the end of your manuscript, and update any in-text citations to match accordingly. Please see our Supporting Information guidelines for more information: http://journals.plos.org/plosone/s/supporting-information. 

Reviewers' comments:

Reviewer's Responses to Questions

**Comments to the Author**

1. Is the manuscript technically sound, and do the data support the conclusions?

Reviewer #1: Yes

Reviewer #2: Partly

2. Has the statistical analysis been performed appropriately and rigorously? 

Reviewer #1: Yes

Reviewer #2: Yes

3. Have the authors made all data underlying the findings in their manuscript fully available?

Reviewer #1: Yes

Reviewer #2: Yes

4. Is the manuscript presented in an intelligible fashion and written in standard English?

Reviewer #1: Yes

Reviewer #2: No

5. Review Comments to the Author

Reviewer #1: The article by Mc Carthy et al address the question of the role of ApoE4 fragment in the toxicity observed previously in vitro. In order to investigate the impact of ApoE4 fragment at the scale of a whole organism, they performed different treatment during zebrafish development. They established a follow up of mortality, developmental defects and neuronal localization.

While an ApoE3 fragment have a limited impact on zebrafish embryos, the ApoE4 fragment treatment results in a severe phenotype on development and survival. As observed in previous microglial cell experiment, ApoE4 fragment tends to localize in the nuclei compartment. The validation of the neurotoxicity hypothesis of ApoE4 fragment would provide a valuable argument regarding Alzheimer disease development. Several points might need some improvements to strengthen the conclusions.

Major questions:

1- The developmental phenotypes seem to reflect a general toxicity rather than organ specific targeting, with edemas, smaller ear, loss of pigmentation. Could the authors check for ApoE4 fragment localization in different organ or cell types like heart or muscle cells? It will help to understand the toxicity development, even more since the embryos did not recover after treatment removal.

2- At the cellular level, the immunolabeling is not homogeneous regarding the quality of staining. Could the author provide lower magnification of the section in addition to the high magnification? It will help to apprehend the results. In addition, the anti-His- labeling is quite different between the SupFig1 and figure 4 for the same 48 hpf stage, could the authors explain this difference?

The nuclear localization is rather clear, but it would add to have a 3D reconstruction of the optical plans to fully convince of the subcellular location.

Another question to strengthen the important result of co-localization of ApoE4 fragment and Tau PHF1: could the authors provide a quantification of this neuronal co-localization? How many cells could be observed for each embryo?

3- The previous data of the authors have been obtained on microglial cells. What about zebrafish microglial / macrophages presence during the treatment? It can be discussed if no data are available.

Minor points-

1-Figure 1 Moderate score should be considered as severe score

2- How many sections per embryos have been observed?

3- Table 1: could the authors provide the reference # of the NeuN antibody, since no mouse polyclonal is found on the catalog.

4- For Touch Escape Response locomotor test, usually the trajectory length and total time of swimming are plotted (Campanari et al 2021 as an example), could the authors provide these parameters? From the supplementary movie, it looks like the embryo is on its back which means it has a vestibular/ear problem. Could the authors have another example of embryo with no abnormality?

Reviewer #2: The manuscript “An amino-terminal fragment of apolipoprotein E4 leads to behavioral deficits, increased PHF-1 immunoreactivity, and mortality in zebrafish” describes the involvement of nApoE41-151 in behavioral deficit and immune hyperreactivity in zebrafish. The study apparently has a good scientific relevance; however, it is not properly conducted and presents a series of serious problems. The main serious points are:

1. My major concern is that the authors have conducted the experiments on 24 hpf, 48hpf and 72hpf zebrafish embryos for modelling a disease like AD. In the introduction section it is mentioned that the majority of AD cases are characterized as late onset AD, only 5% cases comprise of early onset of AD. How can it be concluded if it is AD or any other neurological outcome such as poor outcome of brain injury after treatment of exogenous ApoE ¬fragments or any tauopathy other than AD? Justify. As we know that a proposed hypothesis should be examined from at least three different angles/aspects. There was no genetic manipulation in the present study.

2. An experiment like this requires very strict adherence to randomization protocols, as the potential for bias is high. I suggest that the authors review the ARRIVE and PREPARE guidelines (PLoS Biol. 2020 https://doi.org/10.1371/journal.pbio.3000410; Lab Anim. 2018 Apr;52(2):135-141) to better describe the methodology.

3. Line 75-76…References are required to justify the statement. The introduction portion may be improved by including some recently reported research. Some reference the authors should consider are as follows-

a. Ding Y, Lei L, Lai C, Tang Z. Tau Protein and Zebrafish Models for Tau-Induced Neurodegeneration. J Alzheimers Dis. 2019;69(2):339-353. doi: 10.3233/JAD-180917. PMID: 31006683.

b. Matsui H, Ito J, Matsui N, Uechi T, Onodera O, Kakita A. Cytosolic dsDNA of mitochondrial origin induces cytotoxicity and neurodegeneration in cellular and zebrafish models of Parkinson's disease. Nat Commun. 2021 May 25;12(1):3101. doi: 10.1038/s41467-021-23452-x. PMID: 34035300; PMCID: PMC8149644.

c. Bhattarai P, Cosacak MI, Mashkaryan V, Demir S, Popova SD, Govindarajan N, Brandt K, Zhang Y, Chang W, Ampatzis K, Kizil C. Neuron-glia interaction through Serotonin-BDNF-NGFR axis enables regenerative neurogenesis in Alzheimer's model of adult zebrafish brain. PLoS Biol. 2020 Jan 6;18(1):e3000585. doi: 10.1371/journal.pbio.3000585. PMID: 31905199; PMCID: PMC6964913.

d. Pradhan LK, Sahoo PK, Chauhan S, Das SK. Recent Advances Towards Diagnosis and Therapeutic Fingerprinting for Alzheimer's Disease. J Mol Neurosci. 2022 Jun;72(6):1143-1165. doi: 10.1007/s12031-022-02009-7. Epub 2022 May 12. PMID: 35553375.

e. Shenoy A, Banerjee M, Upadhya A, Bagwe-Parab S, Kaur G. The Brilliance of the Zebrafish Model: Perception on Behavior and Alzheimer's Disease. Front Behav Neurosci. 2022 Jun 13;16:861155. doi: 10.3389/fnbeh.2022.861155. PMID: 35769627; PMCID: PMC9234549.

4. Line 440…the authors have mentioned that… “our results suggest that they may inhibit BMP signaling”… only pigmentation change was reported here. Please explain. Molecular marker-based studies are required to justify the statement.

5. Figure 4 and supplementary figure 1 qualities are not up to mark. Cell boundaries are not distinguishable in some cells.

6. The manuscript should be read by a native English speaker.

6. PLOS authors have the option to publish the peer review history of their article (what does this mean?). If published, this will include your full peer review and any attached files.

Reviewer #1: No

Reviewer #2: **Yes: **Lilesh Kumar Pradhan

---

## [Author Response · Author response to Decision Letter 0]

23 Sep 2022

Response to the Reviewers and Academic Editor: We would like to thank the Reviewers for taking the time to evaluate our manuscript and provide helpful suggestions and comments in improving the manuscript. All changes to the revised document are shown in red, marked up version. We believe the suggestions and comments raised by the Reviewers have significantly improved the quality of the manuscript for which we are grateful.

Article title: An amino-terminal fragment of apolipoprotein E4 leads to behavioral deficits, increased PHF-1 immunoreactivity, and mortality in zebrafish

Comments from the Editor:

1. Please ensure that your manuscript meets PLOS ONE's style requirements, including those for file naming. Response: Using the formatting guidelines provided by PLOS ONE, we have made sure to use the proper style requirements.

2. We note that the grant information you provided in the ‘Funding Information’ and ‘Financial Disclosure’ sections do not match. When you resubmit, please ensure that you provide the correct grant numbers for the awards you received for your study in the ‘Funding Information’ section. Response: We have ensured that the grant information now matches.

3. Please remove any funding-related text from the manuscript and let us know how you would like to update your Funding Statement. Response: We have deleted funding-related text from the manuscript and have updated our Funding Statement.

4. PLOS ONE now requires that authors provide the original uncropped and unadjusted images underlying all blot or gel results reported in a submission’s figures or Supporting Information files. Response: We have no submitted gels or blots as figures in this manuscript, so no raw images are available to submit to the journal.

5. Please include captions for your Supporting Information video files at the end of your manuscript, and update any in-text citations to match accordingly. Response: We have now included captions for our supporting information videos in the revised manuscript and have updated any in-text citations to match accordingly.

Review 1 points of concern: We appreciate the thoroughness by which the Reviewer examined our manuscript and believe the suggestions given have significantly improved the manuscript. In this manner, we attempted to answer every concern of this Reviewer and the additional experimentation recommended by this Reviewer has in our minds strengthened the manuscript.

1. The developmental phenotypes seem to reflect a general toxicity rather than organ specific targeting, with edemas, smaller ear, loss of pigmentation. Could the authors check for ApoE4 fragment localization in different organ or cell types like heart or muscle cells? It will help to understand the toxicity development, even more since the embryos did not recover after treatment removal.

Response: We believe this is an excellent suggestion by the Reviewer and have now incorporated new experimental data showing staining within muscle tissue in the tail region of zebrafish embryos (new Fig 7). In this manner we now show a disruption of muscle cell organization following treatment with nApoE41-151 (herein in designated E4 fragment) that is similar to what has previously been reported in the literature for the overexpression of a mutant AChR in zebrafish (Ref 55 of revised manuscript). As the Reviewer suggests, these findings strengthen our previous results and provide a mechanism as to the motor behavior deficits even after removal of the fragment from the media. These new results are describe in the result section (lines 380-388) and in the discussion (lines 491-497). 

2. At the cellular level, the immunolabeling is not homogeneous regarding the quality of staining. Could the author provide lower magnification of the section in addition to the high magnification? It will help to apprehend the results. In addition, the anti-His- labeling is quite different between the SupFig1 and figure 4 for the same 48 hpf stage, could the authors explain this difference? Response: In response to this concern regarding the quality of staining in our original Fig. 4B, we now present new experimental data showing high magnification of the nuclear labeling of nApoE41-151 following treatment. We believe this data strongly support the nuclear localization of the fragment. Moreover, we now also include new data showing the co-localization of the E4 fragment with a specific nuclear, neuronal marker, NeuN (Fig. 4D). In addition, as suggested by the Reviewer, we now include new experimental data in whole embryo mounted sections at low magnification in order to display overall staining in the entire embryo. These results are presented on lines 303-315 of the revised manuscript. We have also removed the original SupFig1 from the revised manuscript. In general, we consistently observed a more or less punctate pattern of nApoE41-151 labeling that was perfectly consistent to our previous results in BV2 microglial cells (ref 18). The nuclear localization is rather clear, but it would add to have a 3D reconstruction of the optical plans to fully convince of the subcellular location. Response: to answer the Reviewer directly, we did not perform 3D reconstruction of the optical planes to fully show the subcellular location. However, we do have multiple z-stack image analyses that indicate nuclear labeling (data not shown) and we believe our new data shown in Fig 4 showing strong co-localization of the E4 fragment with NeuN, support nuclear localization. Another question to strengthen the important result of co-localization of ApoE4 fragment and Tau PHF1: could the authors provide a quantification of this neuronal co-localization? How many cells could be observed for each embryo? Response: Unfortunately, we did not quantify this potential relationship, but we plan on exploring the phf-1 findings in more detail for a future study.

3. The previous data of the authors have been obtained on microglial cells. What about zebrafish microglial / macrophages presence during the treatment? It can be discussed if no data are available. Response: This is an important point raised by the Reviewer. In fact, during the time-frame of development that we employed zebrafish embryos, it is known that they do not as of yet express fully functional glial cells (Ref 29). This is one reason we focused on neuronal populations.in the current study. Additionally, we have previously identified the E4 fragment within neurons of human postmortem AD brain sections. Therefore, we were also interested in exploring whether we could identify nuclear localization in neurons in this study. We have added language addressing this on lines 232-236 of the revised manuscript.

Minor points

1-Figure 1 Moderate score should be considered as severe score. Response: We agree with the Reviewer on this point and appreciate this observation. We have now amended Fig 1 to reflect this change.

2- How many sections per embryos have been observed? Response: A minimum of 3 sections per embryo were analyzed. This language was added to the revised manuscript on lines 220-222.

3- Table 1: could the authors provide the reference # of the NeuN antibody, since no mouse polyclonal is found on the catalog. Response: We have now added the exact clone (1B87) that was used in our studies in the revised Table 1.

4- For Touch Escape Response locomotor test, usually the trajectory length and total time of swimming are plotted (Campanari et al 2021 as an example), could the authors provide these parameters? Response: We have included new experimental data whereby the total distance traveled is reported (Revised Fig. 6C). We also included heat map data from an averaged control and a low-performing nApoE41-151-treated larvae (Fig 6D). From the supplementary movie, it looks like the embryo is on its back which means it has a vestibular/ear problem. Could the authors have another example of embryo with no abnormality? Response: In one respect, we agree with the Reviewer on this point; however another perspective is what if the unusual behavior of this particular fish was in fact due to the deleterious effects of nApoE41-151 treatment? It is for this reason that we respectfully request to keep the current supplementary video for this sample.

Review 2 points of concern: We appreciate the insightful comments that this Reviewer provided and believe those suggestions have improved the manuscript.

1. My major concern is that the authors have conducted the experiments on 24 hpf, 48hpf and 72hpf zebrafish embryos for modelling a disease like AD. In the introduction section it is mentioned that the majority of AD cases are characterized as late onset AD, only 5% cases comprise of early onset of AD. How can it be concluded if it is AD or any other neurological outcome such as poor outcome of brain injury after treatment of exogenous ApoE ¬fragments or any tauopathy other than AD? Justify. Response: We believe this is an excellent point raised by the Reviewer. The goal of the present study was not to employ zebrafish as a model of AD per se, but to test whether a risk factor associated with AD could lead to toxicity or other potential negative consequences in an in vivo model. In this manner, the use of wild-type zebrafish embryos was used to extend our previous in vitro findings in transformed cells [Ref 16-18]. The primary weakness in our previous published work was the use of transformed, BV2 microglia cells in an entirely in vitro model system. These weaknesses can be summarized as relying on data from a single, murine, immortal microglia cells that may not be representative of normal, non-transformed cells. Another potential caveat of our previous studies was the reliant upon in vitro model systems to investigate the pathophysiological actions of nApoE41-151. Therefore, the primary goal of the current study was to expand our in vitro findings to an intact organism consisting of zebrafish embryos and larvae. Zebrafish have emerged an excellent model organism for studies of vertebrate biology. We have heavily edited the revised manuscript to reflect this idea and to emphasize this was not a model of AD. Our future lab directions are to test the E4 fragment in more appropriate AD models using zebrafish. Changes in the revised manuscript to reflect this new tone given on lines 81-84 of the introduction and lines 424-432 of the discussion.

2. An experiment like this requires very strict adherence to randomization protocols, as the potential for bias is high. I suggest that the authors review the ARRIVE and PREPARE guidelines (PLoS Biol. 2020 https://doi.org/10.1371/journal.pbio.3000410; Lab Anim. 2018 Apr;52(2):135-141) to better describe the methodology. Response: We believe this was an excellent suggestion by the Reviewer and have extensively revised the methods to incorporate the ARRIVE guidelines including how we blinded behavioral studies, average weight, sex etc. as well as providing actual calculations of the numbers of embryos in terms of power of study. We are confident that our methodology was extremely rigorous and our results transparent and reproducible. 

3. Line 75-76…References are required to justify the statement. The introduction portion may be improved by including some recently reported research. Some reference the authors should consider are as follows- Response: We have now provide new references to support the statement in the introduction (Ref 19-21) in the revised manuscript. We have also included the suggested references provided by the Reviewer in the revised manuscript. We thank the Reviewer for providing these important references to us.

4. Line 440…the authors have mentioned that… “our results suggest that they may inhibit BMP signaling”… only pigmentation change was reported here. Please explain. Molecular marker-based studies are required to justify the statement. Response: We agree with the Reviewer on this point and because we have no data to support this statement, all statements regarding BMP signaling have been removed from the revised manuscript.

5. Figure 4 and supplementary figure 1 qualities are not up to mark. Cell boundaries are not distinguishable in some cells. Response: Because the Reviewer had issues with these two figures in terms of quality, we have removed the supplementary Figure from the revised manuscript. In addition, we have completely revised Fig 4 and replaced that data with new data showing crisp, high-magnification images that clearly show nuclear localization of the E4 fragment (revised Fig 4A-C). 

6. The manuscript should be read by a native English speaker. Response: All authors including the corresponding author are native English speakers. We have carefully reviewed the manuscript for any typographical and/or grammatical errors. If the Reviewer could be more specific as to specific areas where our English is subpar, we would be happy to revise those sections.

---

## [Editor Report · Decision Letter 1]

2 Oct 2022

PONE-D-22-19048R1An amino-terminal fragment of apolipoprotein E4 leads to behavioral deficits, increased PHF-1 immunoreactivity, and mortality in zebrafishPLOS ONE

Dear Dr. Rohn,

Thank you for submitting your manuscript to PLOS ONE. Please feel free to further edit your manuscript (as you requested to add more figure(s)). 

We look forward to receiving your revised manuscript.

Kind regards,

Weidong Le

Academic Editor

PLOS ONE

---

## [Author Response · Author response to Decision Letter 1]

6 Oct 2022

Response to the Reviewers and Academic Editor: We would like to thank the Reviewers for taking the time to evaluate our manuscript and provide helpful suggestions and comments in improving the manuscript. All changes to the revised document are shown in the red, marked up version. We believe the suggestions and comments raised by the Reviewers have significantly improved the quality of the manuscript for which we are grateful.

Article title: An amino-terminal fragment of apolipoprotein E4 leads to behavioral deficits, increased PHF-1 immunoreactivity, and mortality in zebrafish

Comments from the Editor:

1. Please ensure that your manuscript meets PLOS ONE's style requirements, including those for file naming. Response: Using the formatting guidelines provided by PLOS ONE, we have made sure to use the proper style requirements to the best of our knowledge.

2. We note that the grant information you provided in the ‘Funding Information’ and ‘Financial Disclosure’ sections do not match. When you resubmit, please ensure that you provide the correct grant numbers for the awards you received for your study in the ‘Funding Information’ section. Response: We have ensured that the grant information now matches.

3. Please remove any funding-related text from the manuscript and let us know how you would like to update your Funding Statement. Response: We have deleted funding-related text from the manuscript and have updated our Funding Statement.

4. PLOS ONE now requires that authors provide the original uncropped and unadjusted images underlying all blot or gel results reported in a submission’s figures or Supporting Information files. Response: We have no submitted gels or blots as figures in this manuscript, so no raw images are available to submit to the journal.

5. Please include captions for your Supporting Information video files at the end of your manuscript, and update any in-text citations to match accordingly. Response: We have now included captions for our supporting information videos in the revised manuscript and have updated any in-text citations to match accordingly.

Review 1 points of concern: We appreciate the thoroughness by which the Reviewer examined our manuscript and believe the suggestions given have significantly improved the manuscript. In this manner, we attempted to answer every concern of this Reviewer and the additional experimentation recommended by this Reviewer has in our minds strengthened the manuscript.

1. The developmental phenotypes seem to reflect a general toxicity rather than organ specific targeting, with edemas, smaller ear, loss of pigmentation. Could the authors check for ApoE4 fragment localization in different organ or cell types like heart or muscle cells? It will help to understand the toxicity development, even more since the embryos did not recover after treatment removal.

Response: We believe this is an excellent suggestion by the Reviewer and have now incorporated new experimental data showing staining within muscle tissue in the tail region of zebrafish embryos (new Fig 7). In this manner we now show a disruption of muscle cell organization following treatment with nApoE41-151 (herein designated the E4 fragment) that is similar to what has previously been reported in the literature for the overexpression of a mutant AChR in zebrafish (Ref 55 of revised manuscript). As the Reviewer suggests, these findings strengthen our previous results and provide a mechanism as to the motor behavior deficits even after removal of the fragment from the media. These new results are described in the result section (lines 393-401) and in the discussion (lines 499-505). 

2. At the cellular level, the immunolabeling is not homogeneous regarding the quality of staining. Could the author provide lower magnification of the section in addition to the high magnification? It will help to apprehend the results. In addition, the anti-His- labeling is quite different between the SupFig1 and figure 4 for the same 48 hpf stage, could the authors explain this difference? Response: In response to this concern regarding the quality of staining in our original Fig. 4B, we now present new experimental data showing high magnification of the nuclear labeling of nApoE41-151 following treatment. We believe this data strongly support the nuclear localization of the fragment. Moreover, we now also include new data showing the co-localization of the E4 fragment with a specific nuclear, neuronal marker, NeuN (Fig. 4D). In addition, as suggested by the Reviewer, we now include new experimental data in whole embryo mounted sections at low magnification in order to display overall staining in the entire embryo. These results are presented on lines 313-325 of the revised manuscript. We have also removed the original SupFig1 from the revised manuscript. In general, we consistently observed a more or less punctate pattern of nApoE41-151 labeling that was perfectly consistent to our previous results in BV2 microglial cells (ref 18). The nuclear localization is rather clear, but it would add to have a 3D reconstruction of the optical plans to fully convince of the subcellular location. Response: to answer the Reviewer directly, we did not perform 3D reconstruction of the optical planes to fully show the subcellular location. However, we do have multiple z-stack image analyses that indicate nuclear labeling (data not shown), and we believe our new data shown in Fig 4 showing strong co-localization of the E4 fragment with NeuN, support nuclear localization. Another question to strengthen the important result of co-localization of ApoE4 fragment and Tau PHF1: could the authors provide a quantification of this neuronal co-localization? How many cells could be observed for each embryo? Response: Unfortunately, we did not quantify this potential relationship, but we plan on exploring the phf-1 findings in more detail in a future study.

3. The previous data of the authors have been obtained on microglial cells. What about zebrafish microglial / macrophages presence during the treatment? It can be discussed if no data are available. Response: This is an important point raised by the Reviewer. In fact, during the time-frame of development that we employed zebrafish embryos, it is known that they do not as of yet express fully functional glial cells (Ref 29). This is one reason we focused on neuronal populations in the current study. Additionally, we have previously identified the E4 fragment within neurons of human postmortem AD brain sections. Therefore, we were also interested in exploring whether we could identify nuclear localization in neurons in this study. We have added language addressing this on lines 232-236 of the revised manuscript.

Minor points

1-Figure 1 Moderate score should be considered as severe score. Response: We agree with the Reviewer on this point and appreciate this observation. We have now amended Fig 1 to reflect this change.

2- How many sections per embryos have been observed? Response: A minimum of 3 sections per embryo were analyzed. This language was added to the revised manuscript on lines 222-223.

3- Table 1: could the authors provide the reference # of the NeuN antibody, since no mouse polyclonal is found on the catalog. Response: We have now added the exact clone (1B87) that was used in our studies in the revised Table 1.

4- For Touch Escape Response locomotor test, usually the trajectory length and total time of swimming are plotted (Campanari et al 2021 as an example), could the authors provide these parameters? Response: We appreciate this excellent recommendation by the Reviewer. We have included new experimental data whereby the total distance traveled, and total time of swimming are reported (Revised Fig. 6). We also included heat map data from a control and a low-performing nApoE41-151-treated larvae (Fig 6C, bottom panels). Interesting, results for these new data, total swimming distance (p=.08) and swimming time (p=.06) just missed significance as compared to untreated controls. From the supplementary movie, it looks like the embryo is on its back which means it has a vestibular/ear problem. Could the authors have another example of embryo with no abnormality? Response: Our anecdotal observations were that many of the E4-treated fish we tested for TEMR exhibited this behavior of being on their back or side. Because we did not validate this finding statistically in the current manuscript, we are not commenting on this finding in the revised manuscript. However, we are providing an additional raw video showing this behavior following treatment with the E4 fragment. 

Review 2 points of concern: We appreciate the insightful comments that this Reviewer provided and believe those suggestions have improved the manuscript.

1. My major concern is that the authors have conducted the experiments on 24 hpf, 48hpf and 72hpf zebrafish embryos for modelling a disease like AD. In the introduction section it is mentioned that the majority of AD cases are characterized as late onset AD, only 5% cases comprise of early onset of AD. How can it be concluded if it is AD or any other neurological outcome such as poor outcome of brain injury after treatment of exogenous ApoE fragments or any tauopathy other than AD? Justify. Response: We believe this is an excellent point raised by the Reviewer. The goal of the present study was not to employ zebrafish as a model of AD per se, but to test whether a risk factor associated with AD could lead to toxicity or other potential negative consequences in an in vivo model. In this manner, the use of wild-type zebrafish embryos was used to extend our previous in vitro findings in transformed cells [Ref 16-18]. The primary weakness in our previous published work was the use of transformed, BV2 microglia cells in an entirely in vitro model system. These weaknesses can be summarized as relying on data from a single, murine, immortal microglia cells that may not be representative of normal, non-transformed cells. Another potential caveat of our previous studies was the reliant upon in vitro model systems to investigate the pathophysiological actions of nApoE41-151. Therefore, the primary goal of the current study was to expand our in vitro findings to an intact organism consisting of zebrafish embryos and larvae. Zebrafish have emerged an excellent model organism for studies of vertebrate biology. We have heavily edited the revised manuscript to reflect this idea and to emphasize this was not a model of AD. Our future lab directions are to test the E4 fragment in more appropriate AD models using zebrafish. Changes in the revised manuscript to reflect this new tone given on lines 82-84 of the introduction and lines 430-439 of the discussion.

2. An experiment like this requires very strict adherence to randomization protocols, as the potential for bias is high. I suggest that the authors review the ARRIVE and PREPARE guidelines (PLoS Biol. 2020 https://doi.org/10.1371/journal.pbio.3000410; Lab Anim. 2018 Apr;52(2):135-141) to better describe the methodology. Response: We believe this was an excellent suggestion by the Reviewer and have now extensively revised the methods to incorporate the ARRIVE guidelines including how we blinded behavioral studies, average weight, sex etc. as well as providing actual calculations of the numbers of embryos in terms of power of study. We are confident that our methodology was extremely rigorous and our results transparent and reproducible. 

3. Line 75-76…References are required to justify the statement. The introduction portion may be improved by including some recently reported research. Some reference the authors should consider are as follows…- Response: We have now provided new references to support the statement in the introduction (Ref 19-21) in the revised manuscript. We have also included the suggested references provided by the Reviewer in the revised manuscript. We thank the Reviewer for providing these important references to us.

4. Line 440…the authors have mentioned that… “our results suggest that they may inhibit BMP signaling”… only pigmentation change was reported here. Please explain. Molecular marker-based studies are required to justify the statement. Response: We agree with the Reviewer on this point and because we have no data to support this statement, all statements regarding BMP signaling have been removed from the revised manuscript.

5. Figure 4 and supplementary figure 1 qualities are not up to mark. Cell boundaries are not distinguishable in some cells. Response: Because the Reviewer had issues with these two figures in terms of quality, we have removed the supplementary Figure from the revised manuscript. In addition, we have completely revised Fig 4 and replaced that data with new data showing crisp, high-magnification images that clearly show nuclear localization of the E4 fragment (revised Fig 4A-C). 

6. The manuscript should be read by a native English speaker. Response: All authors including the corresponding author are native English speakers. We have carefully reviewed the manuscript for any typographical and/or grammatical errors. If the Reviewer could be more specific as to specific areas where our English is subpar, we would be happy to revise those sections.

---

## [Decision Letter · Decision Letter 2]

19 Oct 2022

An amino-terminal fragment of apolipoprotein E4 leads to behavioral deficits, increased PHF-1 immunoreactivity, and mortality in zebrafish

PONE-D-22-19048R2

Dear Dr. Rohn,

We’re pleased to inform you that your manuscript has been judged scientifically suitable for publication and will be formally accepted for publication once it meets all outstanding technical requirements.

Kind regards,

Weidong Le

Academic Editor

PLOS ONE

Additional Editor Comments (optional):

Reviewers' comments:

Reviewer's Responses to Questions

**Comments to the Author**

1. If the authors have adequately addressed your comments raised in a previous round of review and you feel that this manuscript is now acceptable for publication, you may indicate that here to bypass the “Comments to the Author” section, enter your conflict of interest statement in the “Confidential to Editor” section, and submit your "Accept" recommendation.

Reviewer #1: All comments have been addressed

Reviewer #2: All comments have been addressed

2. Is the manuscript technically sound, and do the data support the conclusions?

Reviewer #1: Yes

Reviewer #2: Yes

3. Has the statistical analysis been performed appropriately and rigorously? 

Reviewer #1: Yes

Reviewer #2: Yes

4. Have the authors made all data underlying the findings in their manuscript fully available?

Reviewer #1: Yes

Reviewer #2: Yes

5. Is the manuscript presented in an intelligible fashion and written in standard English?

Reviewer #1: Yes

Reviewer #2: Yes

6. Review Comments to the Author

Reviewer #1: Thank you to the authors for providing additional data and figures.

All comments and questions have been addressed. I have two minor questions:

- Could the authors verify that PHF1 antibody is a mouse polyclonal or a mouse monoclonal antibody?

- Although figure 4 has a better quality and is convincing with the lower magnification image, could the authors provide the separated channel for fig4C: DAPI, His, NeuN, and merge.

Reviewer #2: I note that the author has appropriately addressed all the issues raised. The manuscript is much improved and well-presented.

7. PLOS authors have the option to publish the peer review history of their article (what does this mean?). If published, this will include your full peer review and any attached files.

Reviewer #1: No

Reviewer #2: **Yes: **Lilesh Kumar Pradhan

---

## [Editor Report · Acceptance letter]

7 Dec 2022

PONE-D-22-19048R2 

An amino-terminal fragment of apolipoprotein E4 leads to behavioral deficits, increased PHF-1 immunoreactivity, and mortality in zebrafish 

Dear Dr. Rohn:

I'm pleased to inform you that your manuscript has been deemed suitable for publication in PLOS ONE. Congratulations! Your manuscript is now with our production department. 

Kind regards, 

on behalf of

Dr. Weidong Le 

Academic Editor

PLOS ONE